# The Effect of Propofol versus Inhalation Anesthetics on Survival after Oncological Surgery

**DOI:** 10.3390/jcm11226741

**Published:** 2022-11-14

**Authors:** Laura Jansen, Bente F. H. Dubois, Markus W. Hollmann

**Affiliations:** Department of Anesthesiology, Amsterdam UMC, H1.158 Meibergdreef 9, 1105 AZ Amsterdam, The Netherlands

**Keywords:** sevoflurane, cancer-free survival, oncology, anesthesia, TIVA, volatile anesthetics

## Abstract

Every year, 19.3 million patients worldwide are diagnosed with cancer. Surgical resection represents a major therapeutical option and the vast majority of these patients receive anesthesia. However, despite surgical resection, almost one third of these patients develop local recurrence or distant metastases. Perioperative factors, such as surgical stress and anesthesia technique, have been suggested to play a role to a greater or lesser extent in the development of recurrences, but oncology encompasses a complicated tumor biology of which much is still unknown. The effect of total intravenous anesthesia (TIVA) or volatile anesthesia (VA) on survival after oncological surgery has become a popular topic in recent years. Multiple studies conclude in favor of propofol. Despite the a priori probability that relevant differences in postoperative outcomes are due to the anesthesia technique employed, TIVA or VA, is extremely small. The existing literature includes mainly hypothesis-forming retrospective studies and small randomized trials with many methodological limitations. To date, it is unlikely that use of TIVA or VA affect cancer-free survival days to a clinically relevant extent. This review addresses all relevant studies in the field and provides a substantiated different view on this deeply controversial research topic.

## 1. Introduction

Every year, 19.3 million patients worldwide are diagnosed with cancer [1]. Of these patients, more than half undergo surgical resection. This makes surgical resection of the primary tumor not only one of the most widely used treatments, but also one of the most important therapeutic options for most patients with solid tumors. Of the patients undergoing surgical resection, the vast majority receive anesthesia at least once [2]. Despite resection, approximately one third of these patients develop local recurrence or metastasis. Recurrent or metastatic disease has a dismal prognosis and is associated with the vast majority of cancer-related deaths [3]. In addition to various cancer cell intrinsic properties, environmental factors also play an important role in cancer initiation and progression. Perioperative factors, including surgical stress, pain, and anxiety, as well as the anesthesia technique used during surgery, was supposed to influence the course and progression of cancer directly or indirectly through their impact on tumor-associated environmental factors. In the following paragraphs, we discuss the perioperative vulnerability to the development of tumor recurrences, the plausibility that anesthesia technique affects the likelihood of the development of these recurrences, and the pathophysiology regarding the hypothesized association between the use of some anesthetics and progression of cancer. Finally, we present a summary of the most relevant literature on this issue.

## 2. Perioperative Factors

Several physiological responses occur in the body around surgery: inflammation, increase in circulating catecholamines, immune suppression, and platelet activation. Perioperative triggers might provoke these reactions. One important trigger is stress due to tissue damage from surgical resection. In response to tissue damage, various physiological responses—amongst others, wound healing—occur. This response bears great similarity to pathological responses between tumor cells and the surrounding, nonmalignant cells and non-cellular matrix [4]. Through the interaction of a wound-healing response, existing interactions between cancer cells and their immediate environment can be influenced, with potential consequences on the course (i.e., inhibition or stimulation) of the disease. In addition, pro- and anti-inflammatory responses are induced in an attempt to properly and specifically harness the immune system response in the service of the host. For example, in response to surgical trauma, secretion of various growth factors occurs perioperatively, including vascular endothelial growth factor (VEGF), platelet-derived growth factor (PDGF), and epidermal growth factor (EDF) [5,6,7]. These growth factors stimulate wound healing by activating cellular programs aimed at proliferation and migration of epithelial and endothelial cells, thereby repopulating a tissue and restoring supportive vascular supply. These programs are therefore essential for adequate wound healing. In the context of cancer, however, these programs affect not only the perioperative emergent wound bed, but also the local and systemic context of the tumor [8]. The intrinsically beneficial physiological properties of wound healing are also hijacked by the tumor and harnessed for the benefit of tumor progression. Protective responses can therefore lead to an undesirable, tumor-stimulating effect. This dynamic between cancer cells and their environment is described by the so-called seed-and-soil hypothesis. The hypothesis describes the interaction of malignant cells with their specific context, which can be both stimulating and inhibitory. The formation of metastases through the dissemination and colonization of cancer cells in fertile soil is an important part of this hypothesis [9]. During surgery, dissemination hypothetically occurs through the destruction of not only the tumor, but also the supply and drainage vessels through which tumor cells may enter the circulation [10]. This hypothesis becomes less likely if one takes into account the tumor- doubling time: the time it takes to double the number of tumor cells. When one calculates backwards, in many cases the tumor/metastasis will have existed, undetected, before surgery. The destruction of the vasculature does cause hypoperfusion, resulting in hypoxia and ischemia. This stimulates the expression of hypoxia-induced factors (HIFs), which promote cancer metabolism and tumor growth [11]. Entering the circulation, HIFs interacts with activated platelets, neutrophils, and endothelial cells, as well as transient pro-angiogenic signals. These signals are most likely induced by the surgical inflammatory response and would potentially positively influence formation of metastases [12]. The seed-and-soil hypothesis therefore illustrates the vulnerability to colonization in fertile soil arising from modulation of the immune system and activation or inhibition of neural and/or pro-inflammatory signaling pathways, which in turn prime both the local and systemic environment of the tumor to create a favorable environment for a new metastatic niche [13,14,15]. However, the influence of pro- and/or anti-inflammatory signaling pathways on cancer cell growth is highly heterogeneous and varies by malignancy [14,15]. As a result of the systemic modulation of the immune system, the risk of recurrent and/or metastatic disease is relatively high. The wound-healing response and seed-and-soil hypothesis are important factors in recurrences, but more deserves to be discussed.

Surgical stress from tissue trauma activates more than just a wound-healing response. The stress response (and also anxiety, hypothermia, metabolic disturbances and fasting) activates the sympathetic nervous system, also affecting tumor cells and the microenvironment with possible development of recurrences [16,17,18]. Activation yields increases in circulating catecholamine levels, causing β-adrenoceptor activation. Specifically, the signaling pathway in tumor cells via cyclic adenosine monophosphate (cAMP) is activated, providing upregulation of transcription factors encoding VEGF, MMPs and HIFs, among others [19].

Finally, tissue injury initiates the activation of platelets and tissue factors to promote coagulation. Activation of the coagulation cascade is supposed to influence the formation of metastases because, among other things, platelets contain many cellular growth factors (PDGF, VEGF) as well as matrix proteins and inflammatory mediators [20]. In short, many triggers that activate physiological responses and potentially increase the risk of recurrence exist perioperatively.

Based on previous theory, in 1980, a link between perioperative stress, anesthesia and cancer progression was suggested [21]. In the following years, several retrospective studies were conducted to investigate the association between the type of anesthesia during oncological interventions and cancer-related survival after the surgical procedure. Several retrospective studies suggested that inhalational anesthetics might increase the risk of recurrence and thus negatively affect survival after surgery [22].

## 3. The Hypothesis

In 2016, Wigmore et al. were the first to publish a retrospective study comparing the effect of total intravenous anesthesia (TIVA) versus inhalational anesthetics (IA) on mortality after oncological surgery [22]. The study was conducted in England among more than 7000 patients undergoing resection of the primary tumor. Patients exposed to IA were found to be 1.5 times more likely to die compared with patients who received TIVA (propofol). In the same year, Lee et al. reported that IA compared to TIVA resulted in a significantly higher risk (*p* = 0.037) of developing a recurrence in 300 patients undergoing radical mastectomy for breast cancer [23]. However, no difference was found in mortality. The above studies led to more retrospective reviews, which were finally analyzed in a meta-analysis. As with Wigmore, this meta-analysis showed a difference in mortality to the detriment of IA (Sevoflurane) [24]. Based on the above results, two questions arise: (1) To what extent is the association between the perioperative use of specific anesthetics and the postoperative progression of disease postulated in these retrospective studies biologically plausible? (2) To what extent is the effect reported in these retrospective studies reproducible in methodologically well-conducted randomized trials?

As discussed earlier, tumor biology is highly complex and a subject with much to explore, as well as discuss. Metastasis occurs when cells undergo somatic changes that cause them to spread by infiltrating the lymphatic or vascular network, survive there, and then have the ability to grow at distance [12]. In other words, cancer is complex, with the ultimate course being the result of hundreds of interacting factors. On the other hand, TIVA (e.g., propofol) and IA are agents that are largely similar pharmacodynamically. Both bind to the GABA_A_ receptor and both stimulate the release of neurotransmitters that inhibit the conduction of action potentials in the central nervous system. Despite their similarity, they have been reported to make a fundamental difference (hazard ratio of 1.46 (1.29 to 1.66)) in cancer survival when applied in an oncological context [22]. In addition, it is important to realize that in the development of specific chemo- or immunotherapeutics, an effective agent is said to be present if the therapy gives a 2% to 2.3% reduction in the recurrence rate [25]. Given the hundreds of oncologic factors, the high similarity between the two anesthetics, and the relatively short duration of exposure, a relationship between the two does not seem, a priori, very plausible. Therefore, the a priori probability that the above, short-term variations will have a substantial impact on the course of the disease in an individual patient seems extremely small.

As far as the above is not convincing, it is further supported by a critical review of the static basis of the reported studies. In 2005, Ioannidis published an article in which he emphasizes the importance of a critical review of scientific research [26]. Ioannidis calls attention to the importance of the *p*-value and points out that it is mostly overvalued in the existing literature and the dependence of the a priori probability and the *p*-value receives far too little attention. Ioannidis argues that a study design should take into account three factors: (a) the power of a study; (b) the probability of bias; and (c) the prior probability that a relationship found is real (a priori probability). From this, the positive predictive value is then calculated. The article shows that for most studies, the probability that the outcome matches reality is less than 0.5. Only a correctly conducted randomized controlled trial (RCT) and a meta-analysis of correctly conducted RCTs have a positive predictive value of 0.85. For smaller, less transparent studies, the positive predictive value drops to only 0.20. Furthermore, Ioannidis points out that replicates are essential in studies with an a priori low prior probability.

## 4. Mechanisms of Anesthetic on Cancer

When pathophysiological knowledge substantiates an association found in a study, the likelihood that the association actually exists is greater. The hypothesis underlying the question whether TIVA or IA is superior in the oncologic patient is based on immunosuppressive properties of sevoflurane and protective properties of propofol. Results of studies of these properties are highly variable and, therefore, not a solid basis for the hypothesis.

Although tumor growth, recurrence, and metastasis can be differently affected by the immune status (pro- or anti-inflammatory) at different time points and in different tumor types, the suggested protective properties of propofol could be explained by its anti-inflammatory properties. These properties have been demonstrated in several areas in both animal and in vitro studies. Propofol provides suppression of prostaglandin and cytokine production [27], prevents immunosuppression [28], reduces migration of cancer cells through MMP suppression, and provides increased activity of natural-killer (NK) cells [29]. Furthermore, propofol has been shown to reduce both cancer cell motility and the degree of invasiveness, and lastly gives reduction of HIF-1a [30].

In human studies, it has been shown that more activated T-helper cells circulated and lower concentrations of VEGF-C, TGF-B and IL-6 were found—all markers associated with the formation of angiogenesis and metastases.

IA were reported to have opposite effects which can be traced to known cytoprotective properties in the heart, brain and kidney, as well as reduction in infarct size in models for ischemia-reperfusion injury [31]. The cytoprotective properties are detrimental in oncology, for example, an upregulation of HIF1-α has been demonstrated in vivo [32], also reduced NK cell activity and increased migration of cancer cells [30,33]. In vitro studies support this hypothesis. Thus, based on in vivo and in vitro studies, IA would promote immunosuppression and stimulate a pro-malignant environment. However, contrary to previous findings, sevoflurane has also been shown to reduce cell motility and invasion by reducing MMP2 and MMP9. Given the heterogeneous nature of tumor biology, the in vitro studies should be interpreted with caution. The studies are not directly equivalent to the human cellular environment and therefore cannot be extrapolated one-to-one to clinical outcomes.

## 5. Existing Literature

Interest in the influence of anesthesia technique on oncological outcome measures has increased significantly in recent years. This is partly due to the previously mentioned study by Wigmore et al. [22]. Based on these results, dozens of retrospective studies followed, focused on basic pathophysiology, and producing heterogenous results. These results are summarized in Table 1. In early 2021, a meta-analysis was published by Chang et al. with 19 studies comparing propofol with volatile anesthetics [24]. The primary outcome was mortality and cancer-free survival in patients undergoing surgery for a malignancy. In terms of overall mortality, a difference was found in favor of propofol. However, no such difference was found in the duration of cancer-free survival. The main limitation of this meta-analysis is that it involves only retrospective studies. At most, such studies generate hypotheses, but do not confirm or reject them. For this reason, it is important to check whether there are randomized studies that confirm the hypothesized effect.

Yan et al. published two randomized studies including 80 patients each with breast cancer who underwent breast-conserving resection or radical mastectomy under propofol/remifentanyl TIVA or IA with sevoflurane. The primary outcomes were the concentrations of VEGF, TGF-B in serum, and the expression of myeloid-deriving suppressor [49,50]. As a secondary outcome, both studies looked at mortality and cancer-free survival after two years of follow-up. Both studies showed no significant difference in cancer-free survival or mortality. However, since breast cancer has a relatively good two-year survival, these results are difficult to interpret and a real effect can be neither demonstrated nor excluded on the basis of these data. This problem is compounded by the relatively small groups of patients in the two studies. Guerrero Orriach et al. published a randomized study including 100 patients with infiltrating bladder carcinoma, comparing the effect of general anesthesia in combination with locoregional analgesia or systemic opioids on cancer-related survival after radical cystectomy [57]. A subgroup analysis to the effect of propofol versus sevoflurane showed a difference in favor of propofol (*p* = 0.02).

Similarly, a second subgroup analysis of the effect of propofol combined with an epidural versus sevoflurane combined with opioids showed a difference in favor of propofol (*p* = 0.02). However, the study has several important limitations. First, only a limited number of prognostic characteristics were included, which may unfairly consider both patient groups as equal. A sample size calculation was missing, which makes it unclear on which assumptions the number of included patients is based. It also remains unclear how many patients were included in the subgroup analyses. Finally, as discussed earlier, the a priori chance of a real difference in outcome between patients who underwent resection under propofol or sevoflurane is very small. Although a significant difference in survival between the two study groups was reported, given the very low prior probability, no solid conclusion can be drawn from this finding. To reach a more reliable conclusion, the dichotomous way of thinking should be converted to a more nuanced, continuous way of thinking: how likely is the difference assumed in the hypothesis, what is the effect size, and is the expected effect size clinically relevant? However, the a priori probability of an anesthesia-related effect on survival after oncological surgery is extremely small, and if present, may be very limited and clinically irrelevant.

The latest and most important randomized study was carried out by Sessler et al. [48]. This study was the largest (2108 patients) multicenter (13 hospitals around the world) trial employing high-quality methodology, including women with breast cancer. Patients underwent mastectomy or wide local resection. Propofol and a paravertebral block (the expected most tumor-suppressive anesthesia technique) was compared with sevoflurane and the use of opioids (the expected most tumor-promoting anesthesia technique) with the outcome being cancer-recurrence-free days. The follow-up was five to six years.

The main result was that no difference was found between the two groups (*p* = 0.84). Strictly speaking, the calculated sample size was too small for the observed event rate (number of tumor recurrences). Moreover, the trial ended early as the number of inclusions was not achieved and the futility calculation clearly showed that no difference was to be expected even with further inclusions. It remains for the reader to judge whether one considers a difference clinically relevant if one could not have found it in 2100 patients and not even a signal towards benefit for propofol and regional anesthesia was found.

There are currently several large randomized studies (CAN-, GA-CARES-, VAPOR-C TRIALS etc.) in extensive abdominal surgery comparing propofol with sevoflurane. These studies will have to show that even in operations with more surgical stress, more pain, and more opioid need, no significant difference is going to be found.

A limitation of our paper is that it is not a systematic review and we were not able to statistically pool the results into a meta-analysis. We deliberately chose not to write a systematic review, because we want to show precisely how heterogeneous the outcomes are in the research field and how important it is not only to look at the numerical outcomes of the applied statistics, but precisely also to consider the a priori probability and thus the applied statistics (the alpha and beta error margins chosen among others).

A second limitation is that our search strategy was not classically designed as might be expected of a systematic review. It was not searched by two independent researchers from the start, but only after an initial screening. However, we believe that the chance of missed studies is small because two authors did review the references of the initially included studies.

## 6. Conclusions

Oncology consists of a complex tumor biology of which much is not yet known and not fully understood. In addition to tumor-specific factors, multiple perioperative factors play a role, such as inflammation, pain, stress, and surgical trauma. A popular topic of research came from the question as to whether TIVA or IA is superior in the oncologic patient. The hypothesis is based on immunosuppressive properties of sevoflurane and protective properties of propofol. Important to realize is that the a priori probability that two anesthetics (propofol and inhalational anesthetics) will provide a relevant difference in oncological outcomes is extremely small and that in the context of extraordinary claims requiring extraordinary data, compelling evidence must be put forward to convince the reader that one of the anesthetics can make a significant difference on tumor biology. Therefore, judging the influence of anesthetics on tumor biology is challenging. The existing literature mainly includes many hypothesis-forming retrospective studies; small suboptimal reported randomized trials with many methodological limitations in which the difference found is very unlikely to be existent or clinically relevant. The study by Sessler et al. is leading to date and concludes that no difference is found in the use of propofol or sevoflurane with the outcome measure being cancer-free survival days.

## Figures and Tables

**Table 1 jcm-11-06741-t001:** Studies investigating the effect of intravenous anesthetics versus inhalation anesthetics on overall survival and cancer-free survival.

Author	Overall Survival	Recurrence-Free Days
Schmoch 2021 [34]		
Takeyama 2021 [35]	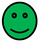	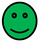
Koo 2020 [36]		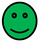
Lai 2020 [37]	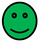	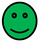
Enlund 2020 [38]	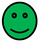	
Huang 2020 [39]	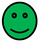	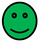
Dong 2020 [40]		
Hong 2019 [41]		
Huang 2019 [42]		
Lai 2019 [43]	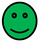	
Sung 2021 [44]		
Yoo 2019 [45]		
Oh 2019 [46]		
Lai 2019 [47]	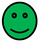	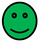
Sessler 2019 [48]		
Yan 2018 [49]		
Yan 2019 [50]		
Zheng 2018 [51]	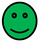	
Wu 2018 [52]	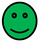	
Oh 2018 [53]		
Kim 2017 [54]		
Jun 2017 [55]	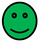	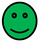
Wigmore 2016 [22]	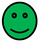	
Lee 2016 [23]		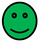
Enlund 2014 [56](Breast cancer)		
Enlund 2014 [56](Colon cancer)		
Enlund 2014 [56](Rectal cancer)		

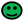
: no significant difference. 
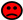
: a significant difference in favor of intravenous anesthestics. 

: outcome measurement not described.

## Data Availability

Not applicable.

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
