# Peer review of "The Effect of Propofol versus Inhalation Anesthetics on Survival after Oncological Surgery"

_jcm, 2022, doi:10.3390/jcm11226741_

Round 1
Reviewer 1 Report
Greetings
I read your article with interest and found it relevant and informative. You have covered the topic well, and the manuscript is well-written. Although I do not have any major comments to make, in my opinion, a few aspects needs attention to improve the information provided
1. In the table, instead of a pictorial depiction of significant benefits, you can put the exact values of the differences found in the studies.
2. It will be interesting to know (include) how you have searched and included the relevant and recent articles.
3. While your conclusion is acceptable, a sentence on the limitation of your article will be informative.
Best of luck
Author Response
We would like to thank the reviewer for the kind remarks and valuable comments on our manuscript. We have taken all comments into consideration and adjusted the manuscript accordingly.
I read your article with interest and found it relevant and informative. You have covered the topic well, and the manuscript is well-written. Although I do not have any major comments to make, in my opinion, a few aspects needs attention to improve the information provided
- In the table, instead of a pictorial depiction of significant benefits, you can put the exact values of the differences found in the studies.
De main message of this figure is to illustrate that results from the various studies are quite randomly distributed suggestive of a lack of a solid effect. We have therefore decided not to display the exact numbers
- It will be interesting to know (include) how you have searched and included the relevant and recent articles.
Initial searches in Pubmed were conducted by one author using the following search terms:
“neoplasia,” “neoplasm,” “tumor,” “cancer,” “malignancy,” “malignant neoplasm,”
AND
“propofol,” “diprivan,” “total intravenous anesthesia,” “target-controlled infusion,” “inhalation anesthetics,” “volatile anesthetics,” “anesthetic gases,” “sevoflurane,” “desflurane,” “isoflurane,” “enflurane,” “halothane,” “balanced anesthesia,” “mortality,”
AND
“all-cause mortality,” “survival,” “overall survival,” “recurrence,” “recurrence-free survival,” “cox regression,” “proportional hazard model,” and “Kaplan-Meier.”
Then two authors reviewed all references of the relevant articles. This is not a systematic review, but a review article in which we have not used sharply delineated inclusion criteria, but have chosen to discuss what we believe to be the most relevant papers most extensively. The table does include all studies comparing intravenous anesthesia and volatile anesthesia on (cancer-free) survival in oncologic surgery.
- While your conclusion is acceptable, a sentence on the limitation of your article will be informative.
We agree with the reviewer's comment. We have added the following sentence (259-269):
‘A limitation of this paper is that it is not a systematic review and we were not able to statistically pool the results into a meta-analysis. We deliberately chose not to write a systematic review, because we want to show precisely how heterogeneous the outcomes are in the research field and how important it is not only to look at the numerical outcomes of the applied statistics, but precisely also to consider the a priori probability and thus the applied statistics (the alpha and beta error margins chosen among others).
A second limitation is that our search strategy was not classically designed as might be expected of a systematic review. It was not searched by two independent researchers from the start, but only after an initial screening, however, we believe that the chance of missed studies is small because two authors did review the references of the initially included studies.’

Reviewer 2 Report
The article is good structured but in conclusion the study of Sessler et al. points out that there is no difference with all the problems of that study. The selection of patients is also a problem, it would be nice to see (in conclusion) what happens with other TIVA techniques and in other oncology patients ie. neurosurgical where TIVA is a predominant technique.
Author Response
We would like to thank the reviewer for the kind remarks and valuable comments on our manuscript. We have taken all comments into consideration and adjusted the manuscript accordingly.
Reviewer 2:
The article is good structured but in conclusion the study of Sessler et al. points out that there is no difference with all the problems of that study. The selection of patients is also a problem, it would be nice to see (in conclusion) what happens with other TIVA techniques and in other oncology patients ie. neurosurgical where TIVA is a predominant technique.
In this review, we focused on the putative difference in cancer-free survival in favor of intravenous anesthesia compared to volatile anesthesia. We fully support that it is interesting to further explore the differences between different TIVA groups. For example, the use of opiates versus loco-regional techniques has also been mentioned and explored several times. Not to mention the full post-operative anesthetic management which could be influential. We have limited ourselves in this case to a small part of the anesthetic management. One purpose of this is to show that it is very unlikely that the choice of TIVA versus volatile anesthesia could make a significant difference in survival in patients in a complete oncologic treatment pathway and randomizing an operative policy does not guarantee that the post-operative analgesic management is the same in both groups.
Unfortunately, we were unable to find any studies in neurosurgery.
